# Association between Low-Grade Chemotherapy-Induced Peripheral Neuropathy (CINP) and Survival in Patients with Metastatic Adenocarcinoma of the Pancreas

**DOI:** 10.3390/jcm10091846

**Published:** 2021-04-23

**Authors:** Martina Catalano, Giuseppe Aprile, Monica Ramello, Raffaele Conca, Roberto Petrioli, Giandomenico Roviello

**Affiliations:** 1School of Human Health Sciences, University of Florence, Largo Brambilla 3, 50134 Florence, Italy; marti_cat@yahoo.it; 2Department of Oncology, San Bortolo General Hospital, AULSS8 Berica, 36100 Vicenza, Italy; giuseppe.aprile@aulss8.veneto.it; 3Oncology Unit, Department of Medical, Surgical & Health Sciences, University of Trieste, Piazza Ospitale, 34100 Trieste, Italy; monica.ramello@asuits.sanita.fvg.it; 4Division of Medical Oncology, Department of Onco-Hematology, IRCCS-CROB, Referral Cancer Center of Basilicata, via Padre Pio 1, 85028 Rionero, Vulture (PZ), Italy; raffaeleconca@hotmail.it; 5Department of Medicine, Surgery and Neurosciences, Medical Oncology Unit, University of Siena, Viale Bracci-Policlinico “Le Scotte”, 53100 Siena, Italy; r.petrioli@ao-siena.toscana.it; 6Department of Health Sciences, University of Florence, Viale Pieraccini 6, 50139 Florence, Italy

**Keywords:** pancreas, neuropathy, taxanes, survival

## Abstract

The combination of nab-paclitaxel and gemcitabine demonstrated greater efficacy than gemcitabine alone but resulted in higher rates of chemotherapy-induced peripheral neuropathy (CINP) in patients with metastatic pancreatic cancer (mPC). We aimed to evaluate the correlation between the development of treatment-related peripheral neuropathy and the efficacy of nab-P/Gem combination in these patients. mPC patients treated with nab-paclitaxel 125 mg/m^2^ and gemcitabine 1000 mg/m^2^ as a first-line therapy were included. Treatment-related adverse events, mainly peripheral neuropathy, were categorized using the National Cancer Institute Common Toxicity Criteria scale, version 4.02. Efficacy outcomes, including overall survival (OS), progression-free survival (PSF), and disease control rate (DCR), were estimated by the Kaplan–Meier model. A total of 153 patients were analyzed; of these, 47 patients (30.7%) developed grade 1–2 neuropathy. PFS was 7 months (95% CI (6–7 months)) for patients with grade 1–2 neuropathy and 6 months (95% CI (5–6 months)) for patients without peripheral neuropathy (*p* = 0.42). Median OS was 13 months (95% CI (10–18 months)) and 10 months (95% CI (8–13 months)) in patients with and without peripheral neuropathy, respectively (*p* = 0.04). DCR was achieved by 83% of patients with grade 1–2 neuropathy and by 58% of patients without neuropathy (*p* = 0.03). In the multivariate analysis, grade 1–2 neuropathy was independently associated with OS (HR 0.65; 95% CI, 0.45–0.98; *p* = 0.03). nab-P/Gem represents an optimal first-line treatment for mPC patients. Among possible treatment-related adverse events, peripheral neuropathy is the most frequent, with different grades and incidence. Our study suggests that patients experiencing CINP may have a more favorable outcome, with a higher disease control rate and prolonged median survival compared to those without neuropathy.

## 1. Introduction

Metastatic pancreatic cancer (mPC) is associated with poor survival rates, with a worldwide 5-year survival rate lower than 5% [1]. Until recently, patients with advanced pancreatic cancer had limited treatment options. Nab-paclitaxel plus gemcitabine (nab-P/Gem) is a first-line treatment option approved in the US and Europe based on the international phase III MPACT trial results for its proven superiority over gemcitabine [2]. The median overall survival (OS) for nab-P/Gem was 8.5 months compared to 6.7 months for gemcitabine (hazard ratio (HR), 0.72; *p* < 0.001), the median progression-free survival (PFS) was 5.5 months versus 3.7 months (HR, 0.69; *p* < 0.001), and the overall response rate (ORR) was 23% and 7%, respectively (*p* < 0.001). An updated report showed a final OS for nab-P/Gem versus Gem alone of 8.7 months and 6.6 months, respectively (HR, 0.72; *p* < 0.001) [3]. Along with its cost-effectiveness in first-line, a number of retrospective series have confirmed nab-P/Gem as an active, effective, well-tolerated, and cost-effective regimen even in pretreated patients [4,5,6,7,8]. Taxanes are microtubule-stabilizing agents (MTSAs), including polyoxyethylated castor oil-based paclitaxel, docetaxel, and ABI-007 (nab-paclitaxel) used for the treatment of various cancers. ABI-007 is a new polyoxyethylated castor oil-free formulation of paclitaxel developed to overcome the limitations attributed to the solvent Kolliphor EL (previously called Cremophor EL) and improve the therapeutic index and safety profile of solvent-based paclitaxel (sb-P) [9]. Peripheral neuropathy (PN) is the major toxicity related to taxanes [6,10,11,12]. Although it is not completely clear how taxanes cause PN, in vitro studies have demonstrated that taxanes interrupt axonal transport mediated by microtubules, leading to neuropathy [13]. Other data demonstrate damage to mitochondria that may underlie a metabolic axonal failure in chemotherapy-induced peripheral neuropathy (CIPN) [14,15]. A novel study using the zebrafish model suggested that paclitaxel-induced neuropathy may depend on interactions between skin nerve endings and epidermal basal keratinocytes through the matrix metalloproteinase MMP-13 [16]. Peripheral neuropathy is a troublesome side effect experienced by many cancer patients that should be actively managed during its course and after the end of treatment [17,18]. This specific toxicity can be dose-limiting and may persist indefinitely in some cases [19,20].

In phase II/III trials of various tumor types, nab-paclitaxel regimens demonstrated improved efficacy and tolerability compared with solvent-based taxanes [21,22]. Recent retrospective studies have reported a 30.4% to 56.8% incidence of CIPN during nab-P/Gem combination chemotherapy [2,5,8]. In the MPACT trial, 54% of the patients experienced any-grade CIPN. Subset analysis of the MPACT trial and the report by Cho et al. demonstrated that the development of treatment-related peripheral neuropathy represents an independent, positive predictive factor of OS [8,23]. Although nab-P/Gem combination therapy is now a consolidated treatment in clinical practice, some reports suggest a possible link with long-term prognosis [16,17]. Data regarding its possible association with survival in the real world are still lacking. The aim of our study is to evaluate the correlation between the development of CINP and the efficacy of nab-P/Gem combination therapy in patients with metastatic pancreatic cancer treated in the real world.

## 2. Materials and Methods

### 2.1. Eligibility Criteria

Patients investigated in this study derive from the multicenter retrospective study NAPA that evaluated patients with metastatic PC treated with first-line nab-P/Gem according to clinical guidelines. This study involved patients treated at 5 Italian oncological units (after the last amendment that has added another center) with first-line nab-P/Gem between January 2015 and December 2018 [24]. We performed a retrospective study of Italian oncological centers across the North, Central, and South of Italy. This study enrolled adult patients with Eastern Cooperative Oncology Group Performance Status (ECOG-PS) 0–1 and histologically confirmed metastatic adenocarcinoma of the pancreas. Patients were required to have adequate hepatic, hematologic, and renal function (including bilirubin level ≤ the upper limit of normal, absolute neutrophil count ≥ 1.5 × 10^9^/L, platelet count ≥ 100,000/mm^3^, and hemoglobin level ≥ 9 g/dL). Patients who had completed surgery or adjuvant treatments (chemotherapy or radiation therapy) for more than 6 months were evaluated. Patients who had at least one cycle of treatment completed were included. Serious cardiovascular problems (i.e., ejection fraction < 40%, myocardial infarction) or infections represented exclusion criteria. The protocol was approved by the Institutional Review Board for clinical trials of Tuscany: Section AREA VASTA CENTRO, number:14565_oss; all patients gave their written consent.

### 2.2. Treatment Schedule and Response Assessments

The initial dose of nab-P/Gem was chosen according to a pivotal study: intravenous infusion of nab-paclitaxel 125 mg/m^2^, followed by gemcitabine 1000 mg/m^2^ administered intravenously on days 1, 8, and 15 every 4 weeks. A second or additional therapy line was administered according to the single-center experience.

Patients received antiemetic medication at the beginning of each treatment cycle and adequate doses of analgesic drugs to provide optimal pain control. Chemotherapeutic cycles were administered with absolute neutrophil count > 1500/μL, hemoglobin ≥ 9 g/dL, and platelets > 100,000/mm^3^, granulocyte-olony stimulating factor (G-CSF) was administered according to the local clinical practice. Clinical, radiological, and biochemical pretreatment assessments were performed within 2 weeks from treatment beginning. Blood tests were performed at baseline and before every single drug administration, while measurement of the carbohydrate antigen (CA)19-9 serum level was performed at baseline and every 12 weeks. Tumor response evaluation was performed every 3 months or earlier when clinically required by spiral computed tomography according to the Response Evaluation Criteria in Solid Tumors (RECIST) version 1.1 [25].

### 2.3. Neuropathy Assessment

Peripheral motor/sensory neuropathy is a disorder characterized by damage or dysfunction of the peripheral motor/sensory nerves. Sensory neuropathy presents as paresthesia, numbness, and pain in the feet and hands [26]. Paresthesia occurs in distal lower extremities with a glove-and-stocking distribution and is most severe on the plantar surface [27]. The severity of most symptoms is mild to moderate, and symptoms generally disappear on cessation of therapy [26,28,29]. Motor neuropathy is usually mild and presents as muscle weakness such as foot drop or difficulty in climbing stairs, decreasing at times fine motor skills [27,30,31]. Neuropathy is graded by subjective complaints of patients and physical examination by clinicians. It was assessed by the National Cancer Institute Common Toxicity Criteria for Adverse Events (NCI-CTCAE) scale, version 4.02 [32]. Grade 1 defines an asymptomatic disorder or loss of deep tendon reflexes or paresthesia where only clinical or diagnostic observations are needed and intervention is not indicated; grade 2 involves moderate symptoms that limit instrumental activities of daily living (ADL); grade 3 involves severe symptoms limiting self-care ADL that need an assistive device; grade 4 involves life-threatening consequences where urgent intervention is indicated. Dose modification, delay, and drug discontinuation related to neuropathy or other adverse events (AE) were performed according to the guidelines.

### 2.4. Statistical Analysis

This study aimed to evaluate whether the development of neuropathy (grade 1–2) positively correlates with the efficacy and survival of patients with metastatic pancreatic cancer treated with nab-P/Gem as first-line treatment. For this purpose, patients were split into two groups: with or without the development of neuropathy. Patient and tumor characteristics plus treatment data were collected as frequency, percentage of categorical variables, median with 95% confidence interval, and range (for continuous variables). Overall survival was evaluated as the time from nab-P/Gem regimen start to death from any cause or the date of the last follow-up visit. Progression-free survival was evaluated as time from treatment initiation to the date of the disease progression. The Kaplan–Meier method with log-rank test was performed to analyze PFS and OS in relation to the development of grade 1–2 neuropathy. The Cox regression model was used to evaluate the prognostic role of neuropathy and other clinical and/or pathological variables. Statistical analysis was performed using STATA software with a statistical significance threshold agreed upon *p* < 0.05.

## 3. Results

### 3.1. Patient Characteristics

From January 2015 to December 2018, 153 patients diagnosed with metastatic PC and treated with first-line nab-P/Gem were retrospectively investigated. Of these, 47 patients (30.7%) developed grade 1–2 neuropathy, and 106 (69.3%) did not develop any neuropathy during treatment. No grade 3 or 4 CIPN was reported. The median age was 67 years (range, 47–84) for the grade 1–2 neuropathy group and 66 years (range, 50–84) for patients without peripheral neuropathy (*p* = 0.8). Eighteen (38.3%) patients with grade 1–2 neuropathy and 37 (34.9%) without neuropathy were 70 or older (*p* = 0.7). Males were more represented in the no peripheral neuropathy group (58.5%) than in patients with PN (55.3 %) without significative differences (*p* = 0.7). Over half of the patients presented with ECOG PS 1 in the two groups (53.2% in patients with CIPN vs. 51.2% in those without CIPN) (*p* = 0.8).

Nineteen (40.4%) patients with neuropathy, and 42 (39.6%) without, presented three or more metastatic sites (*p* = 0.5). Concerning previous treatments, more patients without neuropathy underwent surgery and radiotherapy (11.3% and 25.5%) (*p* = 0.2) than patients with neuropathy (4.3% and 21.3%) (*p* = 0.6) while a biliary stent was previously placed in the 34% and 21.3%, respectively (*p* = 0.2). Basal carbohydrate antigen 19-9 (CA 19-9) levels showed a minimal difference between the two groups (*p* = 0.6). Finally, pain was more present in patients with neuropathy than in the other group (46.8% vs. 34.9%) (*p* = 0.2). Baseline patient characteristics are summarized in Table 1.

### 3.2. Neuropathy and Clinical Outcome

Forty-seven patients (30.7%) developed grade 1–2 neuropathy during treatment. Patients with neuropathy received a mean of six cycles vs. four cycles in patients without neuropathy. Concerning efficacy data, median PFS was 6 months (95% CI (5–6 months)) while median OS was 11 months (95% CI 11 (10–13 months)); no complete responses (CR) were observed, and the disease control rate (DCR) was 66.7% (102 out of 153 patients) among all patients (Table 2).

Patients who developed grade 1–2 neuropathy had a median PFS of 7 months (95% CI (6–7 months)) compared to the PFS of 6 months (95% CI (5–6 months)) for patients without neuropathy (Figure 1, *p* = 0.42).

Meanwhile, OS was 13 months (95% CI (10–18 months)) and 10 months (95% CI (8–13 months)) in patients with and without neuropathy (Figure 2, *p* = 0.04).

DCR was achieved in 83% of patients with grade 1–2 neuropathy and in 58% of patients without neuropathy (*p* = 0.03). The results of the univariate analysis for OS show that age ≥ 70, ECOG-PS 1, number of metastatic sites at baseline ≥ 3, and CA 19–9 ≥ 659 U/mL were found to be negative prognostic factors, whereas previous surgery and grade 1–2 neuropathy (HR: 0.62 95% CI 0.46–0.99, *p* = 0.05) were found to be significantly positive prognostic factors (Table 3).

The multivariate analysis confirms that age ≥ 70, number of metastatic sites, CA 19–9, previous surgery, and grade 1–2 neuropathy were independently associated with OS (Table 3).

## 4. Discussion

The prognosis of metastatic pancreatic cancer is very poor, with an expected median survival of fewer than 12 months and a long-term survival rate of approximately 5%. Chemotherapy is the only feasible treatment and often correlates with even serious adverse events. In MPACT—the randomized trial that established the efficacy of nab-P/Gem combination therapy in patients with advanced stages of disease—peripheral neuropathy was frequently reported as a distressing chemotherapy-related toxicity. A solvent-free form of nab-paclitaxel was developed to reduce taxane-induced neurotoxicity; nevertheless, this side effect still affects more than 50% of all treated patients [33]. CINP remains a major clinical problem in the treatment of these patients that may require chemotherapy dose reduction or cessation, increasing cancer-related morbidity and mortality [34,35]. Moreover, there is a lack of clinical trials focusing on the treatment of established painful CIPN, and duloxetine remains the only treatment with sufficient evidence recommended in the Clinical Practice Guideline from ASCO [36]. The objective of our study was to investigate the independent prognostic role of treatment-related peripheral neuropathy. We evaluated the association between CINP and treatment outcomes and summarized the current knowledge regarding the significance of this correlation. The incidence and course of CIPN vary across different studies. In recent retrospective studies, the incidence of CIPN ranged from 30% to approximately 60%, but it may be underestimated because of the lack of available methods to properly evaluate, report, and grade neurological toxicities. In the MPACT trial, any-grade peripheral neuropathy was reported in 227 patients randomized to the experimental arm (54%); of these, 70 (17%) experienced grade 3 CIPN, but no grade 4 was reported [23]. Dose reduction or treatment discontinuation were required in 10% and 8% of patients with grade 3 CIPN, respectively [2]. In a Korean cohort study, Cho et al. reported CIPN in more than 50% of patients, and 15 (18.5%) of these patients experienced severe grades of toxicity [8]. In the study by You et al., 13 (14.8%) patients developed grade 2 PN, while 16 (18.2%) developed grade 3 PN; 19.3% and 18.2% of all patients needed dose reduction and discontinuation of treatment due to PN, respectively [37]. In our study, 47 patients (30.7%) experienced grade 1–2 CIPN, and of these, 38 (80.8%) required dose reduction, and 30 (44.7%) discontinued treatment. Unlike other retrospective studies in which the incidence of grade 3 neuropathy ranged from 10% to 30% [4,5,27], no grade 3 CIPN was observed in our cohort.

In MPACT, the development of severe CIPN during treatment with nab-P/Gem was associated with longer survival rates (HR 0.55; 95% CI, 0.39–0.79; *p* = 0.0007), and every increase in grade was associated with a 35% reduction in the risk of death (HR, 0.65; 95% CI, 0.58–0.72; *p* < 0.0001). We also reported a significant association between peripheral neuropathy and overall survival. Indeed, in our analysis, patients receiving nab-P/Gem and developing PN had a significant 3-month longer median OS (*p* = 0.04) compared to those who did report any peripheral neurotoxicity. In our study, baseline differences in patient characteristics likely did not play a role in the observed results, and the association between CIPN with survival was confirmed in the multivariate analysis, adjusted for other prognostic factors. Similarly, the results of other studies were in line with our results. Cho et al. highlighted the presence of neurologic adverse events as independent survival prognostic factors (HR 0.302; 95% CI 0.130–0.702, *p* = 0.005) [8]. You et al. reported a significantly longer survival rate in patients with CIPN compared to those without neuropathy in the naive model (10.13 vs. 15.53 months, *p* = 0.007), although this correlation was not confirmed in the landmark model at 6 months, used to reduce lead time bias (11.4 vs. 15.3 months) (*p* = 0.089) [8,23,37]. Various studies of breast cancer have been conducted to identify clinical or molecular risk factors for the development of chemotherapy-induced neuropathy. Older age, hyperglycemia, and obesity or poor nutritional status, such as single-nucleotide polymorphisms (SNPs) in the FGD4, EPHA5, and FZD3 genes and ABCB1 and GSTP1 polymorphisms, have been associated with the development of taxane-related neuropathy [38,39,40,41,42]. Conversely, in pancreatic cancer, data on risk factors for CIPN are still lacking, and the assessment of risk factors as shown for breast cancer should be considered in future studies. Moreover, the association between the neurologic side effects and OS improvement is not yet clear; a relationship could be due to individual drug sensitivity and increased treatment exposure. Indeed, according to Scheithauer et al., patients who are more chemo-responsive might have a better treatment response and simultaneously more adverse events (AEs) [43]. As stated above, the mechanism by which taxanes cause PN is not fully elucidated, although axonal damage has been identified in some studies. Chemotherapy-related cognitive impairment (CRCI) is another side effect of chemotherapy whose etiology is not well identified but appears to be related to impaired white matter integrity [44]. Moreover, neuroinflammation seems to be another possible explanatory mechanism for cognitive impairment and peripheral neuropathy, as highlighted both in clinical and animal studies [45]. Therefore, CINP and CRCI could be contextually assessed in patients treated with these drugs to evaluate a possible correlation between them and highlight their relationship with efficacy outcomes [45,46,47,48].

Although we have demonstrated the favorable impact of CINP on survival, our study has some limitations, mainly owing to its retrospective design, small population sample, and the neuropathy assessment method. Indeed, although the National Cancer Institute’s CTCAE is one of the most widely used clinical tools for detecting neuropathy during chemotherapy, it was not specifically developed to assess pain, is not sensitive to change, and has significant inter-rater variability. Methodologies to assess CIPN in clinical trials have therefore been developed to provide improved evaluation tools and patient-reported outcomes. The EORTC QLQ-CINP is a 20-item quality questionnaire that quantifies symptoms and impairments of sensory, motor, and autonomic neuropathy and has been used in large oncology clinical trials [49]. A more recent methodology, the CIPN-R-ODS, was developed with Rasch analysis to build upon disability scales that provide a linear measurement of CIPN-related disability and will likely be utilized in future CIPN clinical trials [50]. Moreover, the Total Neuropathy Score (TNSc) that incorporates quantitative neurological exams and neurophysiology was recently subjected to Rasch analysis in patients with CIPN and could be used to assess outcomes in future clinical trials [51]. Furthermore, other aspects such as mood, pain, depression, and fatigue should be considered contextually through evaluation systems (e.g., Edmonton Symptom Assessment Screen, MD Anderson Symptom Inventory, or EORTC QLQ30) to assess the impact of neuropathy compared to other symptoms in patients’ daily living.

Currently, available data on the prognostic value of peripheral neuropathy in patients with mPC who are treated with nab-P/Gem are limited. PN can lead to discontinuation of treatment, affecting the overall response to chemotherapy. Considering the positive correlation between PN and efficacy outcomes, the identification of risk factors for CINP and its management becomes critical; indeed, as demonstrated for other drugs (e.g., Tyrosine Kinase Inhibitor, Anti-Epidermal Growth Factor Receptors), the occurrence of adverse events such as hypertension or skin rash could be used as a surrogate of efficacy [52,53]. Moreover, better knowledge on symptom clusters of CIPN may help to improve symptom management in clinical practice [54]. Therefore, a deeper understanding of the exact mechanism of chemotherapy-induced peripheral neuropathy and assessing its correlation with treatment outcomes in large, prospective trials will help to define the role of PN as a possible surrogate marker for the efficacy of chemotherapy.

## 5. Conclusions

The combination of nab-P/Gem demonstrated greater efficacy but higher rates of peripheral neuropathy versus gemcitabine in patients with metastatic pancreatic cancer. Although the incidence of PN is lower in solvent-free forms of taxanes such as nab-paclitaxel, it remains the main problem in combination therapy in pancreatic cancer patients. Despite the limitations of this study, our results suggest a positive correlation between nab-P/Gem therapy response and the development of PN. Treatment-related neuropathy might be a predictor of prognosis in patients with metastatic pancreatic cancer treated with nab-paclitaxel, although prospective large-scale trials are needed to confirm these results.

## Figures and Tables

**Figure 1 jcm-10-01846-f001:**
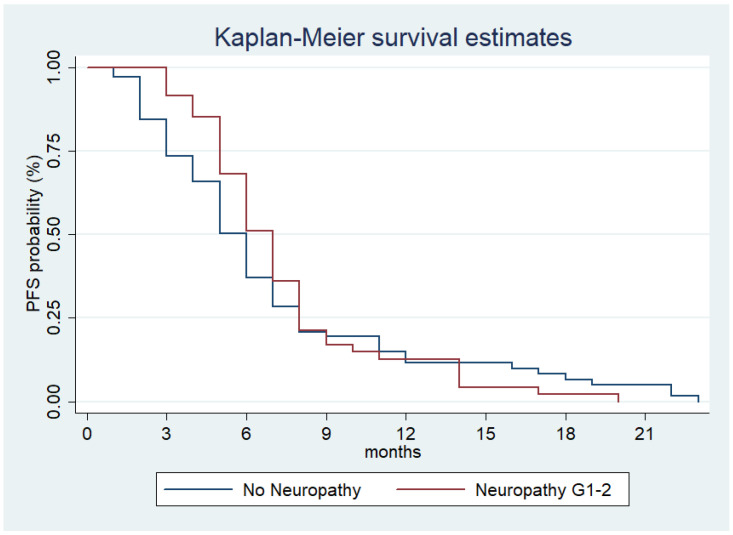
Estimated PFS for nab−Gem according to low-grade CIPN presentation.

**Figure 2 jcm-10-01846-f002:**
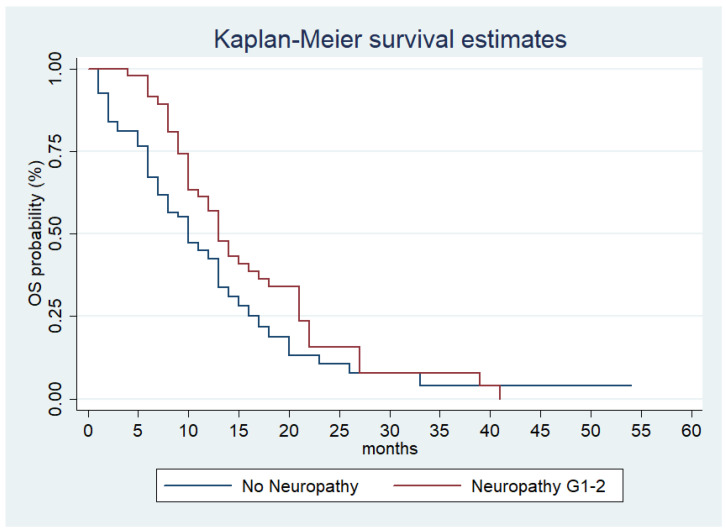
Estimated OS for nab−Gem according to low−grade CIPN presentation.

**Table 1 jcm-10-01846-t001:** Patient characteristics.

	All patients (*n* = 153)	Neuropathy G1–2 (*n* = 47)	No Neuropathy (*n* = 106)	*p*
**Age, years**				
Mean	67	67.5	66	0.8
Range	50–84	47–84	50–84	
≥70	55 (35.9%)	18 (38.3%)	37 (34.9%)	0.7
**ECOG PS**				
1	80 (51.2%)	25 (53.2%)	55 (51.2%)	0.8
**Sex**				
Male	88 (57.5%)	26 (55.3%)	62 (58.5%)	0.7
**Number of metastatic sites**				
≥3	61 (39.9%)	19 (40.4%)	42 (39.6%)	0.5
**Carbohydrate antigen 19-9—U/mL**				
Median	547	401	588	
Range	0.8–700,000	1.8–700,000	0.8–129,718	0.6
**Previous treatment**				
Radiation therapy	14 (9.9%)	2 (4.3%)	12 (11.3%)	0.2
Surgery	37 (24.2%)	10 (21.3%)	27 (25.5%)	0.6
Biliary stent	46 (30.1%)	10 (21.3%)	36 (34%)	0.1
**Pain**				
Yes	59 (38.6%)	22 (46.8%)	37 (34.9%)	0.2

**Table 2 jcm-10-01846-t002:** Best response, PFS, and OS according to neutropenia grade.

	All Patients (*n*= 153)	Neuropathy G1–2 (*n* = 47)	No Neuropathy (*n* = 106)	*p*
PR	58 (37.1%)	16 (34%)	42 (39.6%)	0.03
SD	44 (28.8%)	23 (48.9%)	21 (19.8%)	
DCR (PR + SD)	102 (66.7%)	39 (83%)	63 (58%)	
PD	42 (27.4%)	8 (17%)	34 (32.1%)	
NE	9 (5.9%)	0	9 (8.5%)	
**PFS**				0.42
M-months	6	7	6	
(95% IC)	(5–6)	(6–7)	(5–6)	
**OS**				0.04
M-months	11	13	10	
95% IC	(10–13)	(10–18)	(8–13)	
**Cycles**				0.03
Median	5	6	4	
Range	1–17	2–17	1–17	
Delayed	51 (33.5%)	19 (41.3%)	32 (30.2%)	0.2
Interruption	51 (33.5%)	11 (23.4%)	40 (37.7%)	0.1
Dose reduction	88 (57.5%)	38 (80.8%)	50 (47.2%)	0.01

**Table 3 jcm-10-01846-t003:** Univariate and multivariate analysis for OS.

	HR	IC 95%	*p*
**Univariate**			
**Age ≥ 70**	**1.91**	**1.23–2.90**	**0.004**
**ECOG PS (1 vs. 0)**	**1.45**	**1.22–3.18**	**0.05**
Sex (male vs. female)	1.20	0.79–1.83	0.48
**N. of metastatic sites ≥ 3**	**4.48**	**2.54–8.09**	**<0.001**
**Carbohydrate antigen 19-9 ≥ 659 U/mL**	**1.68**	**1.10–3.22**	**0.01**
Previous radiation therapy	0.47	0.19–1.16	0.1
**Previous surgery**	**0.78**	**0.49–0.98**	**0.04**
Previous biliary stent	0.84	0.54–1.32	0.4
Pain present	1.50	0.98–2.29	0.06
**Neuropathy**	**0.68**	**0.46–0.99**	**0.05**
**Multivariate**			
**Age ≥ 70**	1.53	1.05–2.21	0.03
ECOG PS (1 vs. 0)	1.24	0.85–1.80	0.26
**N. of metastatic sites ≥ 3**	3.98	2.22–5.66	<0.001
**Carbohydrate antigen 19-9 ≥ 659 U/mL**	1.99	1.38–2.88	<0.001
**Previous surgery**	0.51	0.32–0.82	0.006
**Neuropathy**	0.65	0.45–0.98	0.03

## Data Availability

The data used to support the findings of this study are available from the corresponding author upon reasonable request.

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
