# Peer review of "Association between Low-Grade Chemotherapy-Induced Peripheral Neuropathy (CINP) and Survival in Patients with Metastatic Adenocarcinoma of the Pancreas"

_jcm, 2021, doi:10.3390/jcm10091846_

Round 1
Reviewer 1 Report
Neuropathy is somewhat difficult to express as an objective indicator. Although evaluated using the NCI scale, the patient's condition and surgical extent, tumor size, or supportive care, etc are all different, so the evidence for supporting the usefulness of neuropathy is so weak. Nevertheless, it is a good study.
However, it is necessary to present evidence or possibilities to support the authors' conclusions (correlation between neuropathy and response).
Author Response
Neuropathy is somewhat difficult to express as an objective indicator. Although evaluated using the NCI scale, the patient's condition and surgical extent, tumor size, or supportive care, etc are all different, so the evidence for supporting the usefulness of neuropathy is so weak. Nevertheless, it is a good study.
However, it is necessary to present evidence or possibilities to support the authors' conclusions (correlation between neuropathy and response).
Thank you for this positive comment, we revised the discussion adding evidences from literature that report results similar to our study.
Reviewer 2 Report
The authors in the current study have established a correlation between the development of CINP and the administration of Gem/NabPaclitaxel. They have used appropriate study design and methodology for their study.
Author Response
The authors in the current study have established a correlation between the development of CINP and the administration of Gem/NabPaclitaxel. They have used appropriate study design and methodology for their study.
Thank you for this comment.
Reviewer 3 Report
There are many grammatical errors/typos (e.g. missing or superfluous spaces) throughout the text that should be addressed on the following lines: 24, 35, 53, 57, 58, 59, 64, 72, 83, 111 (should be subsection 2.3), 122, 123 (ADL should be defined), 153, 178, 191, 200, 201, 207, 209, 211 (refs), 218, 226, 228, 229, 230, 233, 234, 236, 238 and 257.
In terms of discussion, do the authors think CINP could be used as a surrogate marker?
Can the authors speculate on differences in PFS between patients with and without CINP?
Fig. 2 shows longer OS for patients with no neuropathy but in the discussion the authors highlight a paper from You et al. reporting longer survival in patients with CIPN. How do the authors reconcile these differences?
In Table 1 there is a typo in the Carbohydrate antigen box on the left.
Author Response
There are many grammatical errors/typos (e.g. missing or superfluous spaces) throughout the text that should be addressed on the following lines: 24, 35, 53, 57, 58, 59, 64, 72, 83, 111 (should be subsection 2.3), 122, 123 (ADL should be defined), 153, 178, 191, 200, 201, 207, 209, 211 (refs), 218, 226, 228, 229, 230, 233, 234, 236, 238 and 257.
Thank you for this comment, the typos and errors have been revised.
In terms of discussion, do the authors think CINP could be used as a surrogate marker?
Yes, we suggested this possibility on the discussion.
Fig. 2 shows longer OS for patients with no neuropathy but in the discussion the authors highlight a paper from You et al. reporting longer survival in patients with CIPN. How do the authors reconcile these differences?
There is a mistake in the interpretation of the figure, as we reported a longer survival in patients with CIPN, this result is in line with the literature.
In Table 1 there is a typo in the Carbohydrate antigen box on the left.
We revised the typos
Reviewer 4 Report
Review for Association between low-grade chemotherapy induced peripheral neuropathy (CINP) and survival in patients with metastatic adenocarcinoma of the pancreas.
I enjoyed reading this manuscript, found it very interesting.
Comments:
Abstract, line 18:“greater efficacy” not defined.
Abstract lines 25: Spell out OS, PSF and DCR.
OS: Overall Survival
PSF: progression free survival
DCR: (I don’t read what that is until the Results - disease control rate…. )
Abstract: line 35: “low-grade” (vs high grade or vs. NONE?). Needs to be made clearer.
p.2 line 52: Why is Microtubule capitalized?
Method/assessment , page 2-3 description. What do clinical assessments entail? Are any other patient-reported outcomes done? Specifically, what about cognition, mood, pain, depression, fatigue, etc.? Perhaps like an Edmonton Symptom Assessment Screen, MDAnderson Symptom Inventory, or EORTC QLQ30? If none, this is a limitation/suggestion for further research, to determine if peripheral neuropathy is ‘special’ vs other patient-reported outcomes.
Results.
Results line 151: How can 55% have neuropathy and 59% not? This needs to be corrected or, if accurate, made clearer.
Discussion - currently organized as two large blocks of text. I suggest further subdivision for more paragraphs.
e.g. limitations its own paragraph. (as just one example).
First section of Discussion emphasizes quality of life, but results are set up to emphasis cancer control and survival. There is a disconnect. I suggest the authors either revise results to reflect interest in QOL or revise this first paragraph of the discussion to first report on results previously described.
It seems to me that much of lines 186-197 in Discussion might better belong in Introduction.
Issues that occurred to me, mostly in intro and discussion, to consider addressing in Intro/discussion:
From what I understand CIPN is more likely in taller people. (CITATIONS) Was this found in this case, and was height adjusted for in the different groups? Also the fact that males were less likely is inconsistent with this - again, literature suggests is more common in males. Worth some discussion.
Given focus on likely axonal damage as cause of peripheral neuropathy what of relationship with “chemobrain” - which also has evidence of impairing functions related to white matter. This could serve to expand this paper, appeal to wider audience, and make suggestions. For instance, people can assess for both CIPN and cognitive symptoms and look for relationships with OS, etc. This is relevant, as authors mentioned, other cancers are treated with similar class of chemotherapy agents. References to include with strong cases for chemotherapy-induced cognitive impairment: Seruga et al 2008 Nature Cancer Reviews; and Bernstein et al 2017 meta-analysis Neuroscience and Biobehavioral Reviews.
Assuming the results found here are reliable and would hold in larger sample, I think the manuscript would be strengthened if there were suggestions for why and/or how to to this relationship and what evidence would support one over another. For example, is having peripheral neuropathy a sort of sign that that person is more ‘sensitive’ chemotherapy, and in that regard that person the chemo would ‘work better’ in terms of being anti-cancer properties? Versus it’s that those patients tended to have more chemo (no dose reduction) - so really it’s an indication of having more chemo, so better cancer outcomes? and if an “early indicator of better outcomes — why? And if so, would other more patient-reported outcomes also be an indicator? e.g., chemobrain severity? or fatigue severity? as example… While I appreciate the authors sticking closely to the correlational data, they are allowed, and even encouraged to make some speculation for different models or explanations or hypotheses for why such a relations may exist. This will strengthen, push research further along, Related questions/issues for discussion: which of these would suggest this relationship would be special for pancreas cancer vs. other types? Also, why only low-grade? where does high-grade fit in? or are there cases when chemo dose was reduced, so too much of a confound/limiting factor ? (for discussion).
Some additional references to consider, in addition to those already mentioned above:
Chemotherapy-induced peripheral neuropathy (CIPN) as a predictor of decreased quality of life and cognitive impairment in testicular germ cell tumor survivors. M Chovanec, D Galikova, L Vasilkova, Va De Angelis, K Rejlekova, J Obertova, Z Sycova-Mila, PPalacka, Katarina Kalavska, D Svetlovska, B Mladosievicova, J Mardiak, and M Mego Journal of Clinical Oncology 2020.
another on mechanisms, linking CIPN and and other patient-reported outcomes/symptoms/problems: Vichaya et al. Mechanisms of chemotherapy-induced behavioral toxicities. Frontiers in Neuroscience 2015
Wang, M., Cheng, H.L., Lopez, V. et al. Redefining chemotherapy-induced peripheral neuropathy through symptom cluster analysis and patient-reported outcome data over time. BMC Cancer 19, 1151 (2019). https://doi.org/10.1186/s12885-019-6352-3
Did authors look at other known risk factors of CIPN? E.g., nutritional deficiencies and/or weight? See for example, JPNS 2018. Risk factors for the development of paclitaxel‐induced neuropathy in breast cancer patients. Robertson, et al. and if not, is this limitation/suggestion/recommendation for future work.

Author Response
Comments:
Abstract, line 18:“greater efficacy” not defined.
Abstract lines 25: Spell out OS, PSF and DCR.
OS: Overall Survival
PSF: progression free survival
DCR: (I don’t read what that is until the Results - disease control rate…. )
Abstract: line 35: “low-grade” (vs high grade or vs. NONE?). Needs to be made clearer.
All the typos have been revised in the abstract, we changed the abstract at line 35 to make it more clear.
p.2 line 52: Why is Microtubule capitalized?
It has been a typo, we changed it.
Method/assessment , page 2-3 description. What do clinical assessments entail? Are any other patient-reported outcomes done? Specifically, what about cognition, mood, pain, depression, fatigue, etc.? Perhaps like an Edmonton Symptom Assessment Screen, MDAnderson Symptom Inventory, or EORTC QLQ30? If none, this is a limitation/suggestion for further research, to determine if peripheral neuropathy is ‘special’ vs other patient-reported outcomes.
Thank you for this comment, unfortunately we did nor perform all that analysis, we reported this on the limitation of the study.
Results.
Results line 151: How can 55% have neuropathy and 59% not? This needs to be corrected or, if accurate, made clearer.
Thank you for this comment, we correct the data.
Discussion - currently organized as two large blocks of text. I suggest further subdivision for more paragraphs.
We divided the discussion in 5 paragraphs.
e.g. limitations its own paragraph. (as just one example).
Thank you, we added this paragraph.
First section of Discussion emphasizes quality of life, but results are set up to emphasis cancer control and survival. There is a disconnect. I suggest the authors either revise results to reflect interest in QOL or revise this first paragraph of the discussion to first report on results previously described.
Thank you for this comment, we revised this paragraph emphasizing efficacy outcome data instead of quality of life not analysed in our work.
It seems to me that much of lines 186-197 in Discussion might better belong in Introduction.
We revised this part of discussion.
Issues that occurred to me, mostly in intro and discussion, to consider addressing in Intro/discussion:
From what I understand CIPN is more likely in taller people. (CITATIONS) Was this found in this case, and was height adjusted for in the different groups? Also the fact that males were less likely is inconsistent with this - again, literature suggests is more common in males. Worth some discussion.
Thank you for this comment, unfortunately we did not evaluate the height aspect in our cohort. Regarding gender instead, accidentally the data reported in the text was incorrect and has been edited (see table 1). Therefore, our results agree with the data reported in the literature.
Given focus on likely axonal damage as cause of peripheral neuropathy what of relationship with “chemobrain” - which also has evidence of impairing functions related to white matter. This could serve to expand this paper, appeal to wider audience, and make suggestions. For instance, people can assess for both CIPN and cognitive symptoms and look for relationships with OS, etc. This is relevant, as authors mentioned, other cancers are treated with similar class of chemotherapy agents. References to include with strong cases for chemotherapy-induced cognitive impairment: Seruga et al 2008 Nature Cancer Reviews; and Bernstein et al 2017 meta-analysis Neuroscience and Biobehavioral Reviews.
Thank you for these interesting comments and tips. Although we did not delve into the subject in our study, hint for possible future studies has been added.
Assuming the results found here are reliable and would hold in larger sample, I think the manuscript would be strengthened if there were suggestions for why and/or how to this relationship and what evidence would support one over another. For example, is having peripheral neuropathy a sort of sign that that person is more ‘sensitive’ chemotherapy, and in that regard that person the chemo would ‘work better’ in terms of being anti-cancer properties? Versus it’s that those patients tended to have more chemo (no dose reduction) - so really it’s an indication of having more chemo, so better cancer outcomes? and if an “early indicator of better outcomes — why? And if so, would other more patient-reported outcomes also be an indicator? e.g., chemobrain severity? or fatigue severity? as example… While I appreciate the authors sticking closely to the correlational data, they are allowed, and even encouraged to make some speculation for different models or explanations or hypotheses for why such a relations may exist. This will strengthen, push research further along, Related questions/issues for discussion: which of these would suggest this relationship would be special for pancreas cancer vs. other types? Also, why only low-grade? where does high-grade fit in? or are there cases when chemo dose was reduced, so too much of a confound/limiting factor ?
According your suggestion, we revised the discussion highlighting the questions that you presented, in addition, we added as possible future studies in the limitation of the study.
Some additional references to consider, in addition to those already mentioned above:
Chemotherapy-induced peripheral neuropathy (CIPN) as a predictor of decreased quality of life and cognitive impairment in testicular germ cell tumor survivors. M Chovanec, D Galikova, L Vasilkova, Va De Angelis, K Rejlekova, J Obertova, Z Sycova-Mila, PPalacka, Katarina Kalavska, D Svetlovska, B Mladosievicova, J Mardiak, and M Mego Journal of Clinical Oncology 2020.
This references was added in the additional section on relationship between CINP and CRCI
another on mechanisms, linking CIPN and and other patient-reported outcomes/symptoms/problems: Vichaya et al. Mechanisms of chemotherapy-induced behavioral toxicities. Frontiers in Neuroscience 2015
As suggest, a reminder on common mechanisms for CINP and other symptoms (specifically CRCI) was added.
Wang, M., Cheng, H.L., Lopez, V. et al. Redefining chemotherapy-induced peripheral neuropathy through symptom cluster analysis and patient-reported outcome data over time. BMC Cancer 19, 1151 (2019). https://doi.org/10.1186/s12885-019-6352-3
This reference has been added
Did authors look at other known risk factors of CIPN? E.g., nutritional deficiencies and/or weight? See for example, JPNS 2018. Risk factors for the development of paclitaxel‐induced neuropathy in breast cancer patients. Robertson, et al. and if not, is this limitation/suggestion/recommendation for future work.
We have added a recommendation for future studies regarding risk factors for CINP in pancreatic cancer in the section where we talked about the risk factors for CINP in other types of cancer (mainly breast cancer).
Round 2
Reviewer 1 Report
Authors cannot figure out the scientific and practical significance of CINP.
Reviewer 4 Report
The authors have revised the manuscript according to the comments made by the reviewers. For suggestions they did not implement, they adequately explained their reasons.